# Effect of Tibolone on Bone Mineral Density in Postmenopausal Women: Systematic Review and Meta-Analysis

**DOI:** 10.3390/biology10030211

**Published:** 2021-03-10

**Authors:** Lizett Castrejón-Delgado, Osvaldo D. Castelán-Martínez, Patricia Clark, Juan Garduño-Espinosa, Víctor Manuel Mendoza-Núñez, Martha A. Sánchez-Rodríguez

**Affiliations:** 1Research Unit on Gerontology, Facultad de Estudios Superiores Zaragoza, Universidad Nacional Autónoma de México, Mexico City 09230, Mexico; lizettcastrejon@hotmail.com (L.C.-D.); mendovic@unam.mx (V.M.M.-N.); 2Clinical Pharmacology Laboratory, Facultad de Estudios Superiores Zaragoza, Universidad Nacional Autónoma de México, Mexico City 09230, Mexico; castelan@unam.mx; 3Clinical Epidemiology Research Unit, Hospital Infantil de México Federico Gómez, Mexico City 06720, Mexico; osteoclark@gmail.com; 4Research Department, Hospital Infantil de México Federico Gómez, Mexico City 06720, Mexico; juan.gardunoe@gmail.com

**Keywords:** tibolone, lumbar spine, femoral neck, total hip, postmenopausal women, estrogen therapy

## Abstract

**Simple Summary:**

Low bone mineral density (osteoporosis) is associated with vertebral and nonvertebral fractures in postmenopausal women. Tibolone is a low-risk hormone replacement therapy alternative to estrogen therapy, effective in the treatment of menopausal symptoms and prevention of bone loss, but the evidence is controversial. This systematic review with meta-analysis summarizes the clinical trials of the tibolone effect on percentage change of bone mineral density in the lumbar spine, femoral neck, and total hip in postmenopausal women. The results show that tibolone 2.5 mg dose increases the percent change in bone mineral density compared with non-active controls at 24 months in lumbar spine and femoral neck, regardless of the scanner used to evaluate the bone mineral density. No difference was observed when 2.5 mg tibolone dose was compared with estrogen therapy at 24 months, and both treatments have a positive effect on the bone mineral density. In conclusion, tibolone increases bone mineral density compared to non-active controls, and there was no difference when it is compared to estrogenic therapy; thus, tibolone is an alternative treatment for menopausal symptoms and bone protection.

**Abstract:**

Low bone mineral density (BMD) on postmenopausal women causes bone fragility and fracture risk. Tibolone seems to prevent bone loss. Therefore, this systematic review with meta-analysis synthesizes the tibolone effect on BMD percent change in lumbar spine (LS), femoral neck (FN), and total hip (TH) in postmenopausal women. Controlled trials that provided tibolone evidence on the efficacy of tibolone in preventing loss of BMD were included. Regarding the included studies, a pooled mean difference (MD) with 95% confidence intervals (95%CI) was estimated to determine the BMD percentage change. Eleven studies were identified and eight were included in the quantitative analysis. Tibolone at a dose of 2.5 mg increased BMD compared with non-active controls at 24 months in LS (MD 4.87%, 95%CI: 4.16–5.57, and MD 7.35%, 95%CI: 2.68–12.01); and FN (MD 4.85%, 95%CI: 1.55–8.15, and 4.21%, 95%CI: 2.99–5.42), with Hologic and Lunar scanners, respectively. No difference was observed when tibolone 2.5 mg dose was compared with estrogen therapy (ET) at 24 months, LS (MD −0.58%, 95%CI: −3.77–2.60), FN (MD −0.29%, 95%CI: −1.37–0.79), and TH (MD −0.12%, 95%CI: −2.28–2.53). Therefore, tibolone increases BMD in LS and FN compared to non-active controls, and there was no showed difference with ET.

## 1. Introduction

Osteoporosis is characterized by a reduction in bone mass and micro-architectural deterioration of bone tissue, that cause an increase in bone fragility and susceptibility to fracture risk. In postmenopausal women, osteoporosis is the most frequent primary form of the pathology observed after the fifth decade due to bone loss caused by estrogen deficiency that increases bone turnover with an imbalance between bone formation and resorption [1,2].

Different treatment options are recommended in postmenopausal women with osteoporosis, such as selective estrogen receptor modulators (SERMs), bisphosphonates, peptides of the parathyroid hormone family, denosumab, romozumab and other pharmacological intervention to prevent bone loss, like hormone therapy (HT) [1].

HT prevents the accelerated bone turnover and bone loss at all skeletal sites and is considered effective to prevent postmenopausal osteoporosis and reduce the risk of vertebral and non-vertebral fractures regardless of baseline bone mineral density (BMD) [1]. Furthermore, HT alleviates bothersome vasomotor symptoms, menopausal genitourinary syndrome and related issues including impaired sleep, irritability and reduced quality of life [3]. However, HT has some contraindications as estrogen is sensitive to breast, endometrial cancer and adverse effects like weight gain, bloating [3] and unwanted bleeding [4].

Otherwise, tibolone, a synthetic steroid with a structure different from estrogens and SERMs [5], acts differently in distinct tissues and organs and has been classified as a selective tissue estrogenic activity regulator (STEAR) [6]. After absorption, tibolone is metabolized in different tissues, producing estrogenically active metabolites that stimulate estrogen, progesterone, and androgen receptors, with an effect on bone preserving or increasing BMD [5,6]. In addition, tibolone’s clinical efficacy is similar to conventional HT, without stimulating breast and endometrium [7], and unscheduled bleeding is lower than that induced by HT [8].

Tibolone is a treatment for climacteric complaints; moreover, it could be a therapeutic option to prevent BMD loss and consequently reduce the risk of fractures in postmenopausal women. Therefore, this systematic review with meta-analysis was performed to (1) summarize the effect of tibolone on BMD change in lumbar spine, femoral neck, and total hip in postmenopausal women; (2) assess the quality of identified trials with current available tools; and (3) evaluate the safety of tibolone therapy.

## 2. Materials and Methods

This research was designed and reported in accordance with PRISMA guidelines [9], and the protocol was prospectively registered on PROSPERO (CRD42020155956).

### 2.1. Search Strategy

A systematic literature search was conducted in the Cochrane Central Register of Controlled Trials (CENTRAL), MEDLINE, ScienceDirect, Scopus, Epistemonikos, Lilacs, SciELO, IMBIOMED and Medigraphic databases. In addition, a grey literature search was conducted in ScienceDirect conference abstracts, Scopus conference paper, Proquest and TESIUNAM. Searches were carried out in July 2020. In the search, we did not restrict the language and it was limited to studies in humans. The search strategy was built using the following MeSH terms and keywords: “tibolone”, “bone density”, “bone”, “osteoporosis” or “osteoporosis postmenopausal”. The strategy was adapted to each of the databases (Appendix A).

### 2.2. Study Selection

Two reviewers (L.C. and O.D.C.) independently assessed all titles and abstracts to identify studies that potentially meet the eligibility criteria. The reviewers were blinded to each other’s decisions. Disagreements were resolved through discussion and consensus with one additional review author (M.A.S.). Studies were included if they met the following criteria: (1) randomized or non-randomized controlled trial; (2) participants were postmenopausal women, defined as women with a surgical or natural menopause with at least 12 months of amenorrhea; (3) intervention was tibolone at any dose compared to non-active controls (placebo, no treatment or calcium therapy) or an active treatment with estrogens or combined hormone therapy; (4) outcome measured as the BMD percent change in lumbar spine (LS), femoral neck (FN) and total hip (TH); (5) BMD determination was performed by quantitative computed tomography (QCT) or by dual energy X-ray absorptiometry (DXA) Lunar or Hologic densitometers; and (6) follow-up at 12, 24 or 36 months. Studies that included women with unilateral or bilateral oophorectomy, osteoporosis diagnosis, previous fractures, and breast cancer, were excluded.

### 2.3. Data Collection

Eligible studies were reviewed by two authors, data extraction was performed by one investigator (L.C.) and then checked independently for accuracy by the other investigator (O.D.C.). The following data were extracted: first author, year of publication, study design, characteristics of participants, interventions doses, follow-up, type of comparator, bone mineral density results and adverse events (AE). For BMD, the following data were extracted: the mean percent change, standard deviation, and number of participants in both the tibolone and comparator group. Where the standard error of the mean percent change (SEM) was only reported, standard deviation (SD) was calculated with the formula: SD = SEM × square root (n), where n is the number of subjects. When the percent change was reported in figures to accurately measure the mean and standard deviation in the graphs, the image was enlarged to interpolate the data with a drawn line. If necessary, the authors were contacted via e-mail for more information about the study design and results.

### 2.4. Risk of Bias Assessment

Two review authors independently evaluated the risk of bias in the included studies. For randomized studies, the recommendations of the Cochrane risk of bias tool was used [10]; each category was graded as having a low, high or unclear risk of bias. Semi and non-randomized studies were evaluated with the ROBINS-I tool [11], the domains were classified as low, moderate, serious or critical risk of bias. Discrepancies regarding extraction of quantitative data or risk of bias assessment of the studies were resolved by an additional review author.

### 2.5. Statistical Methods

Meta-analysis was performed using Review Manager software version 5.3 from the Cochrane Collaboration [12]. For the analysis, two comparisons were performed, in the first one, the effect of tibolone on the BMD percent change was compared with non-active controls, in this comparison a subanalysis of tibolone dose was included at 12 and 24 months. In the second, the effect of tibolone 2.5 mg dose on the BMD percent change was compared with estrogens, a subanalysis for the follow-up at 36 months was included. Analysis for Hologic and Lunar densitometers were considered. The means of the BMD percent change of each study for LS, FN and TH were analyzed using the inverse variance method to obtain a pooled mean difference (MD) with 95% confidence intervals (95%CI). In the analysis of adverse events, the results are shown as relative risks (RR) with 95%CI. All analyses were conducted in a random-effect approach. Studies with insufficient information to be included in the meta-analysis were excluded for the quantitative analyses. The safety of tibolone was evaluated considering the risk ratio of adverse events reported. For the heterogeneity assessment, I^2^ statistics were performed, considered not important if the value is 0% to 40%, as moderate between 30% to 60%, as substantial if the value is 50% to 90%, and 75% to 100% represents considerable heterogeneity [13]. Publication bias was assessed using the Egger regression asymmetry test when the comparison had at least 10 studies, asymmetry is shown using funnel plots. Leave-one-out sensitivity analysis was conducted to determine if the pooled effects were robust, the study with higher risk of bias was excluded. Sensitivity analysis was carried out only if three or more studies were included in the comparison.

## 3. Results

### 3.1. Literature Search and Study Characteristics

A total of 696 citations was found, the study selection process is illustrated in Figure 1. After removing duplicates, 472 studies remained, and we identified 35 potentially relevant studies by screening titles and abstracts. Once the complete texts had been reviewed, we excluded 24 studies for detailed reasons showed in the Appendix A (Appendix A), the main causes of exclusion were observational studies, participants with previous fractures or bilateral oophorectomy. Eleven studies fulfil the inclusion criteria of systematic review, characteristics of the included studies are summarized in Table 1. Three studies were excluded from quantitative analysis, two due to insufficient data and one because it could not be combined. Finally, eight studies were included in the meta-analysis.

A total of 1529 participants were included in 11 studies. The mean age of women in most of the selected studies was between 49 and 55 years [14,15,16,17,18,19,20,21,22,23], in one trial the mean age of participants was 66.8 years [24]. The mean time since the menopause ranged from 1.2 to 19 years. In concordance with the STRAW + 10 system (Stages of Reproductive Aging Work Shop) [25], nine studies included women with early postmenopause, one study included women with late postmenopause and one study did not report the postmenopausal years of their participants.

**Table 1 biology-10-00211-t001:** Characteristics of the included studies (n = 11).

Author, Year	Study Design	Tibolone (Dose)	n	Comparator (Dose)	n	Follow-Up, Years	Age, Years	PO, Year	Assessment of BMD	Measurement Site	Included in the Meta-Analysia
Rymer et al., 1994 [14]	Non-randomized	2.5 mg	46	No medication	45	2	49.5	1.7	DXA, Hologic	Lumbar spine (L1–L4) Femoral neck	Yes
Berning et al., 1996 [15]	Controlled double-blind randomized	2.5 mg 1.25 mg	33 34	Placebo	23	2	52.1	1.8	Quantitative computed tomography	Lumbar spine (L1–L3)	No
Bjarnason et al., 1996 [24]	Controlled double-blind randomized	2.5 mg + 400 mg Ca 1.25 mg + 400 mg Ca	28 28	Placebo + 400 mg Ca	13	2	66.8	19	DXA, Hologic	Lumbar spine (L2–L4)	Yes
Lippuner et al., 1997 [16]	Open semi-randomized	2.5 mg	30	Estradiol 2 mg + DYD Transdermal estradiol patch 50 μg + DYD No medication	28 26 31	2	--	--	DXA, Hologic	Lumbar spine (L2–L4) Femoral neck	Yes
Thiebaud et al., 1999 [17]	Randomized	2.5 mg	16	CEE 0.625 mg + MPA	20	3	54.2	1.2	DXA, Hologic	Lumbar spine Femoral neck Total hip	Yes
Beardsworth et al., 1999 [18]	Controlled randomized	2.5 mg	22	No treatment	20	2	53.2	5.2	DXA, Lunar	Lumbar spine (L2–L4) Femoral neck	No
CasteloBranco et al., 2000 [19]	Open semi-randomized	2.5 mg	23	Estradiol valerate 4 mg + androgens Transdermal estradiol patch 50 μg No treatment	23 26 24	1	53.1	2.7	DXA, Lunar	Lumbar spine	Yes
Milner et al., 2000 [20]	Open semi-randomized	2.5 mg	14	CEE 0.625 mg plus norgestrel No treatment	18 38	2	53.9	4.5	DXA, Lunar	Lumbar spine (L1–L4) Femoral neck	No
Gallagher et al., 2001 [21]	Controlled double-blind randomized	0.3 mg + 500 mg Ca 0.625 mg + 500 mg Ca 1.25 mg + 500 mg Ca 2.5 mg + 500 mg Ca	132 136 127 131	Placebo + 500 mg Ca	130	2	52.4	2.5	DXA, Hologic and Lunar	Lumbar spine (L1–L4) Femoral neck Total hip	Yes
Roux et al., 2002 [22]	Double blind, randomized	2.5 mg + 500 mg Ca 1.25 mg + 500 mg Ca	70 73	Estradiol 2 mg + NETA + 500 mg Ca	68	2	54.1	3.9	DXA, Hologic	Lumbar spine (L2–L4), Femoral neck Total hip	Yes
Gambacciani et al., 2004 [23]	Open randomized	2.5 mg + 1 g Ca 1.25 mg + 1 g Ca	15 27	Ca 1 g	11	2	52.7	2.3	DXA, Lunar	Lumbar spine (L2–L4) Femoral neck	Yes

PO: Post-menopause, mean time since menopause; BMD: bone mineral density; DXA: Dual energy X-ray absorptiometry; Ca: Calcium; DYD: Dydrogesterone; CEE: Conjugated equine estrogens; MPA: Medroxyprogesterone acetate; NETA: Norethindrone acetate.

Four studies were non- or semi-randomized [14,16,19,20], while seven were randomized [15,17,18,21,22,23,24]. A tibolone 2.5 mg dose was administered in 11 studies, 1.25 mg dose in 5 studies, while doses of 0.3 mg and 0.625 mg were administered in one study. Some studies included more than one comparator: placebo/no-treatment or calcium in 9 studies, estrogens in 5 studies (daily doses of conjugated equine estrogens 0.625 mg and estradiol 2 mg or 4 mg) and a transdermal estradiol patch in 2 studies. Five studies determined BMD using DXA Hologic, 4 studies used DXA Lunar, 1 study used both DXA instruments, and 1 study used QCT. Eleven studies reported results of the BMD percent change in LS, 8 studies reported results in FN and 3 studies reported results in TH.

### 3.2. Risk of Bias Assessment in Randomized Trials

In most of the randomized studies in the systematic assessment of bias an unclear or high risk of bias was observed with respect to allocation concealment, blinding of outcome assessment and selective reporting bias (Figure 2a,b).

Regarding the random sequence generation and allocation concealment (selection bias), a low risk of bias was estimated in three studies because their participants were allocated by a random system. Three studies classified by an unclear risk of bias had no information about the generation of a random sequence and one did not specify the block selection process. Besides, all the selected studies were accounted with an unclear risk of bias because they did not provide information about allocation concealment.

With respect to blinding of participants and personnel (performance bias) and blinding of outcome assessment (detection bias), a low risk of bias was considered in four studies double-blind; unclear risk of bias was estimated in two studies without information about blinding participants and/or personnel; and one study, rated with a high risk of bias, was an open study. Moreover, all studies were classified with an unclear risk of bias, if they did not provide enough information for blinding of outcome assessment.

With relation to completeness of follow up (attrition bias), there were dropouts in four studies, the intention to treat analysis was applied so that we considered a low risk of bias. In contrast, three studies describe the dropouts, but there is no comparison with the total number of participants assigned to the groups, which we rated with an unclear risk of bias.

On selective reporting (reporting bias), an unclear risk of bias was considered in all studies. They pre-specified the outcomes reported, according to the procedures, and it is mentioned that the equipment was calibrated, but it is not clear whether it was by the same operator who carried out the measurements, or whether the operators were standardized or certified by the International Society for Clinical Densitometry (ISCD).

Additionally, in the domain of other potential sources of bias, most of the selected studies were rated with a low risk of bias. Finally, most of the randomized studies were considered with an unclear risk of bias.

### 3.3. Risk of Bias Assessment in Non-Randomized Trials

In non-randomized studies, a moderate or serious risk of bias was observed. In the domains addressing issues before the start of the interventions (confounding and selection of participants in the study) and in the domains covering issues after the start of interventions (bias due to deviation from an intended interventions and selection of the reported result) (Figure 2c,d).

Regarding the bias due to confounding, two studies were considered with moderate risk, women with hormone treatment were randomly assigned to the treatment groups. Two studies were classified with serious risk; in one, women were able to choose their treatment and in the other, the baseline variables were different.

Bias in the selection of study participants. The selection may have been related to the intervention and the outcome; three studies were classified with moderate risk as the authors used appropriate methods to adjust for the selection bias and one study with severe risk, as this could not be adjusted for in the analyses.

Bias in classification of interventions: a low risk of bias was considered in three studies with the intervention status well-defined, and one study with some aspects of the assignments of intervention status determined retrospectively was considered to have a moderate risk of bias.

Bias due to deviations from intended interventions: concerning the effect of assignment to intervention, and the effects of starting and adhering to interventions, there were deviations from the usual practice and deviations from the intended interventions, respectively; but the effect on the outcome is expected to be slight; therefore, three studies were classified as being at moderate risk. In one study, the analysis was not appropriate to estimate the effect of adhering to intervention.

Bias due to missing data. Three studies with proportions of missing participants were similar in the intervention groups, they were classified as being at low risk. In one study, a moderate risk of bias was considered because the proportions of missing participants do not differ considerably across intervention groups and the analysis possibly could not have removed the risk of bias arising from the missing data.

Bias in measurement of outcomes. A low risk bias in all studies was considered, and methods used to evaluate intervention group outcomes were comparable, with the possibility of the influence of the knowledge of interventions types on the participants.

Concerning bias in the selection of the reported results, all studies were classified with a moderate risk of bias, and the outcome measurements and analyses are consistent with an a priori plan. Finally, most of the studies present one domain with a serious risk of bias; therefore, the overall risk of bias judgement for non-randomized studies is serious.

### 3.4. Tibolone Compared with Non-Active Controls

In the LS, tibolone 2.5 mg dose [14,16,19,21,24] and 1.25 mg dose [21,24] at 12 months showed an increase in BMD for Hologic and Lunar densitometers. The effect of tibolone on BMD was greater at 24 months with 2.5 mg dose, MD 4.87%, 95%CI: 4.16 to 5.57, 328 participants; I^2^ = 0%, *p* = 0.81; and MD 7.35%, 95%CI: 2.68 to 12.01, 152 participants; I^2^ = 68%, *p* = 0.08, for Hologic and Lunar densitometers, respectively (Table 2). In the FN, tibolone 2.5 mg [14,16,21] and 1.25 mg [21] doses increase BMD at 12 months for both densitometers. Likewise, tibolone showed an increase in BMD at 24 months, especially with 2.5 mg dose, MD 4.85%, 95%CI: 1.55 to 8.15, 287 participants; I^2^ = 96%, *p* < 0.00001, and MD 4.21%, 95%CI: 2.99 to 5.42, 152 participants; I^2^ = 0%, *p* = 0.65, for Hologic and Lunar scanners, respectively (Table 2). No study had a 36-month follow-up or determined BMD in total hip and the funnel plot could not be conducted. In this comparison, sensitivity analysis was performed with a Hologic densitometer and 2.5 mg tibolone dose. Excluding the study with higher risk of bias [14], this analysis does not show a substantial change in the LS and FN. In the LS at 12 months, BMD increases (MD 2.68%, 95%CI: 2.08 to 3.29, 237 participants; I^2^ = 0%, *p* = 0.44); likewise, the BMD percentage change increases after 24 months (MD 4.75%, 95%CI, 3.95 to 5.55, 237 participants; I^2^ = 0%, *p* = 0.75). With regard to FN, the results at 12 and 24 months were MD 1.81%, 95%CI: 0.79 to 2.82, 196 participants; I^2^ = 0%, *p* = 0.93, and MD 3.34% 95%CI: 1.81 to 4.87, 196 participants; I^2^ = 40%, *p* = 0.19, respectively.

### 3.5. Tibolone Compared with Estrogens

The BMD in LS measured with a Hologic densitometer, at 12, 24 and 36 months using tibolone 2.5 mg versus diverse estrogens doses (conjugated equine estrogens (CEE) 0.625 mg [17] and estradiol 2 mg [16,22]), show no difference in the percent change, MD −0.57%, 95%CI: −2.09 to 0.94, 500 participants; I^2^ = 80%, *p* < 0.0001 (Table 2). In FN, there are no studies with Lunar densitometer, using a Hologic scanner there is no difference in the percentage change when comparing 2.5 mg of tibolone with different estrogen doses (CEE 0.625 mg [17] and estradiol 2 mg [16,22]) at 12, 24 and 36 months, MD −0.24%, 95%CI: −1.12 to 0.65, 362 participants; I^2^ = 15%, *p* = 0.32 (Figure 3). Moreover, according to TH, the studies comparing tibolone 2.5 mg and two estrogen doses (CEE 0.625 mg [17] and estradiol 2 mg [22]) at 24 months show no differences in the BMD percent change (Table 2). The sensitivity analysis was carried out with Hologic densitometer and 2.5 mg tibolone dose, after excluding the study with a higher risk of bias [16]. This analysis confirmed the robustness of our findings at 12 and 24 months in LS, MD −1.02%, 95%CI: −3.95 to 1.91, 174 participants; I^2^ = 83%, *p* = 0.02, and MD −0.58%, 95%CI: −5.84 to 4.68, 174 participants; I^2^ = 94%, *p* < 0.0001, respectively. In FN at 24 months, the BMD change shows as results MD −0.19%, 95%CI: −1.98 to 1.60, 174 participants; I^2^ = 45%, *p* = 0.18. The funnel plot and sensitivity analysis for total hip could not be performed.

### 3.6. Adverse Events

Women assigned to the tibolone group were at higher risk of vaginal bleeding than those with non-active controls [14,21,24], RR 2.66, 95%CI: 1.30 to 5.45, 807 participants; I^2^ = 0%, *p* = 0.96 (Figure 4a). On the contrary, tibolone reduced hot flashes, especially with 2.5 mg dose [21], RR 0.29, 95%CI: 0.11 to 0.76, 302 participants. Besides, tibolone was related to anxiety when was compared with non-active controls, with a greater effect at 1.25 mg dose [21], RR 4.52, 95%CI: 1.32 to 15.39, in 303 participants. In comparison with estrogens, tibolone was associated with a lower rate of vaginal bleeding [16,22], RR 0.32, 95%CI: 0.19 to 0.52, in 357 participants; I^2^ = 0%, *p* = 0.88 (Figure 4b), and with a lower rate of breast pain, 2.5 mg dose shows as results RR 0.12, 95%CI: 0.03 to 0.48, in 149 participants [22], and using 1.25 mg dose the results were RR 0.11, 95%CI: 0.03 to 0.48, in 150 participants [22] (Appendix A). Other adverse events for both comparisons are described in the Appendix A.

## 4. Discussion

The present systematic review and meta-analysis synthesizes the evidence of the effect of tibolone in BMD in the lumbar spine, femoral neck, and total hip in postmenopausal women. Concerning that a Cochrane systematic review evaluates the effectiveness and safety of tibolone treatment in postmenopausal and perimenopausal women, most of the included studies reported the effect on vasomotor symptoms (VMS), and five of them had different objectives such as bone loss or fracture prevention, without describing the effect on BMD [8]. Besides, there are two systematic reviews with meta-analysis in the literature regarding the effect of tibolone on bone at 24 months. The first one, published in 2001, analyzed two trials using a Hologic scanner in early postmenopausal women, and showed that a tibolone 2.5 mg dose, compared with placebo, is capable of increasing spinal and femoral BMD [26]. The second, published in 2003, showed that tibolone appears to be as effective at BMD changes as regimens containing any estrogen, using different densitometers [27]. This meta-analysis’ strengths include information with accurate data on the effect size of tibolone over BMD, evaluation at 12, 24 months and when possible at 36 months. In addition, this study explored the risk of bias, a sensitivity analysis, heterogeneity and summarized the evidence for adverse effects published in the trials.

In this study, most of the available data were derived from a comparison with non-active controls. For both densitometers, the increase in BMD is observed for 12 months, with the percent change being greater at 24 months in the lumbar spine and femoral neck, especially with a 2.5 mg tibolone dose. At menopause, both trabecular (cancellous) and cortical bone may be affected, but loss of trabecular bone is more clearly associated with the abrupt decline of the ovarian function at menopause. Trabecular bone consists of 20% of the total bone, which is in the flat bones and in the ends of long bones, and it has ten times the surface/volume ratio of cortical bone. Sex steroids contribute to maintain bone mass mainly by decreasing osteoclastic bone resorption in the trabecular bone, and in this way it suppresses the rate of bone remodeling [28]. In this meta-analysis, the antiresorptive effect of tibolone is observed, and the sensitivity analysis showed a decreased in the heterogeneity of the femoral neck studies; the differences between the results can be explained by the non-randomized study [14] excluded from this analysis. The published meta-analysis, with the Hologic scanner, reports in early-postmenopausal women that 2.5 mg dose of Tibolone at 24 months increases the BMD lumbar spine (MD 5.5%, 95%CI: 4.4 to 6.7, 147 subjects) in data superior to our analysis, and in the femoral neck (MD 4.6%, 95%CI: 3.0 to 6.2, 147 subjects) [26], gaining similar results to this study. On the other hand, bone density measurements from different devices cannot be directly compared with the meta-analysis. In this study, measurements with the Lunar densitometer are greater than those with the Hologic densitometer, in line with this, the reported evidence shows that Hologic spine BMD is typically 11.7% lower than the GE-Lunar BMD [29,30].

According to the comparison between tibolone and estrogens, the information about the 1.25 mg tibolone dose compared with estrogen therapy is scarce [22]. The most frequently used dose in the included studies is 2.5 mg with the Hologic scanner. In our research, at 12, 24 and 36 moths, tibolone 2.5 mg and different estrogen doses in the lumbar spine and femoral neck showed no difference between treatments, in total hip at 24 months, suggesting that tibolone is as effective as estrogen therapy. Similarly, findings of a previous meta-analysis concluded that regarding BMD changes after 2 years of treatment, there is no difference between any estrogen and tibolone [27]. Our results demonstrate that the tibolone effect appears to have a greater effect at the lumbar spine compared with femoral neck and total hip. In addition, BMD increase is observed through time, being greater at 36 months, this trend is observed in the three sites. Heterogeneity in the three measurement sites is probably because of the different estrogen doses.

Consistently, tibolone is an antiresorptive drug, a synthetic steroid analogue of the progestin, norethynodrel, and structurally different from estradiol and SERMs, with unique tissue-specific effects. Tibolone influences the synthesis and metabolism of estrogens, progesterone and endogenous androgens; for example, in the breast it regulates enzymes, in the endometrium the metabolism is tissue-selective, and in bones it acts via activation of estrogen receptors [5,31]. The unique structure of this drug determines its pharmacokinetics, allowing its oral administration once daily. Tibolone is metabolized in the gastrointestinal tract and liver, and its molecular products have different properties: estrogenic (3alpha and 3beta hydroxytibolone) and progestogenic/androgenic (delta^4^ tibolone). About 80% of the total oral dose of tibolone circulates as an inactive sulfated form (3alpha and 3beta sulfated tibolone), then in locally tissues, the sulfatase enzymes desulfated the metabolites into active estrogenic molecules [31]. These metabolites avoid estrogenic stimulation of the breast, inhibiting the sulfatase enzyme and provoking apoptosis; likewise, metabolized progestin prevents the stimulation of the endometrium [7].

In contrast, VMS described by women as hot flashes or night sweats represent the most bothersome symptoms of menopause and the most common reason women seek medical care [32]. Besides, women do not perceive bone demineralization as a negative aspect, until it manifests clinically with a fracture. Regarding this, 45% of 50-year-old women with postmenopausal osteoporosis will suffer fractures of spine, hip, proximal humerus or forearm in the next 10 years; however, 96% of these fractures could occur in women without osteoporosis [33]. The decreased bone strength in osteoporosis predisposes an increased risk of fracture; therefore, bone strength = bone mineral density + bone quality [34].

In this way, a meta-regression of published trials concluded that greater improvements in DXA-based BMD is strongly associated with greater reduction in fracture risk, particularly for spine and hip fractures. For example, if tibolone increases the BMD at 2% of the lumbar spine, it could be associated with a 28% reduction in spine fracture or 22% hip fracture, whereas 4% improvement in femoral neck could be associated with a 55% reduction in spine fracture or 32% hip fracture, according to the meta-regression published [35]. Meanwhile, a network meta-analysis has demonstrated that tibolone is effective for preventing vertebral and non-vertebral fractures [36].

In addition, our analysis supports that tibolone is associated with a lower rate of vaginal bleeding compared to estrogen therapy, and it is suggested that if 47% of women taking combined hormone therapy experience unscheduled bleeding, between 18% and 27% of women taking tibolone will do so [8]. A study has demonstrated that tibolone improved persistent bleeding and breast discomfort after switching from estrogen treatment [37]. Moreover, the evidence suggests that tibolone is more effective than placebo, but less effective than estrogen therapy in reducing VMS [8]. Finally, in this analysis, the relationship between tibolone and anxiety is different to other studies [38,39].

In women with a history of breast cancer, tibolone increase the risk of cancer recurrence and, in women over 60 years of age, it may increase the risk of a stroke. Concerning other long-term adverse events, there is no evidence that tibolone increases the risk or that it differs from estrogen therapy with respect to long-term safety [8]. In relatively healthy postmenopausal women using combined continuous estrogen treatment for one year, the risk of a heart attack and the risk of venous thrombosis increases with longer use. Estrogen therapy also increases the risk of a stroke, breast cancer, gallbladder disease and death from lung cancer [40]; furthermore, common estrogen adverse effects include breast tenderness, bloating, and uterine bleeding [32].

This analysis focused on preventing bone decline with tibolone; however decision-making also incorporates side effects and other measures to improve bone health such as aging, appropriate physical activity, lifestyle, environmental factors, good nutrition, adequate intake of calcium and vitamin D. It is possible to prevent osteoporosis and therefore avoid the intervention of antiresorptive medications or stimulators of bone formation to treat this disease, such as bisphosphonates, demosumab, romosozumab, teriparatide and abaloparatide [36]. Considering the balance between the benefits and risks of tibolone, in addition to using tibolone for postmenopausal symptoms, it is useful for improving BMD.

The findings of this systematic review suggest that tibolone is useful to prevent the decrease in BMD. However, some limitations need to be considered when interpreting the results of this study. First, the evidence from this research cannot be extrapolated to women with osteoporosis or previous fractures, because this population has a greater increase in BMD in response to antiresorptive agents. Second, there are no recent studies, and much of the evidence included was unclear or had a high risk of bias in more than one domain. Third, according to the ISCD, the region of interest in spine BMD measurement is L1–L4. Finally, the comparison and pooling of Hologic and Lunar BMD values is difficult. To solve this problem, raw BMD could be standardized with equations; however, the percentage difference between these two systems could be reduced but not eliminated.

The strengths of this systematic review over meta-analysis is that it is the only one that reports a percentage change in BMD with current available tools and that assesses the quality of identified trials. The specific data of the mean difference in percentage change in BMD could be useful in monitoring bone health, in addition to the possible prediction of fractures. Moreover, the evidence from this systematic review may be valuable in clinical decision-making to treat bothersome menopausal symptoms, with the benefit of bone loss prevention.

## 5. Conclusions

The present meta-analysis provides quantitative data of tibolone improvement of BMD percentage change and demonstrates no difference with estrogen therapy. Quality evidence was unclear or had a serious risk of bias; in addition, tibolone has been shown to have fewer adverse events than estrogen therapy.

## Figures and Tables

**Figure 1 biology-10-00211-f001:**
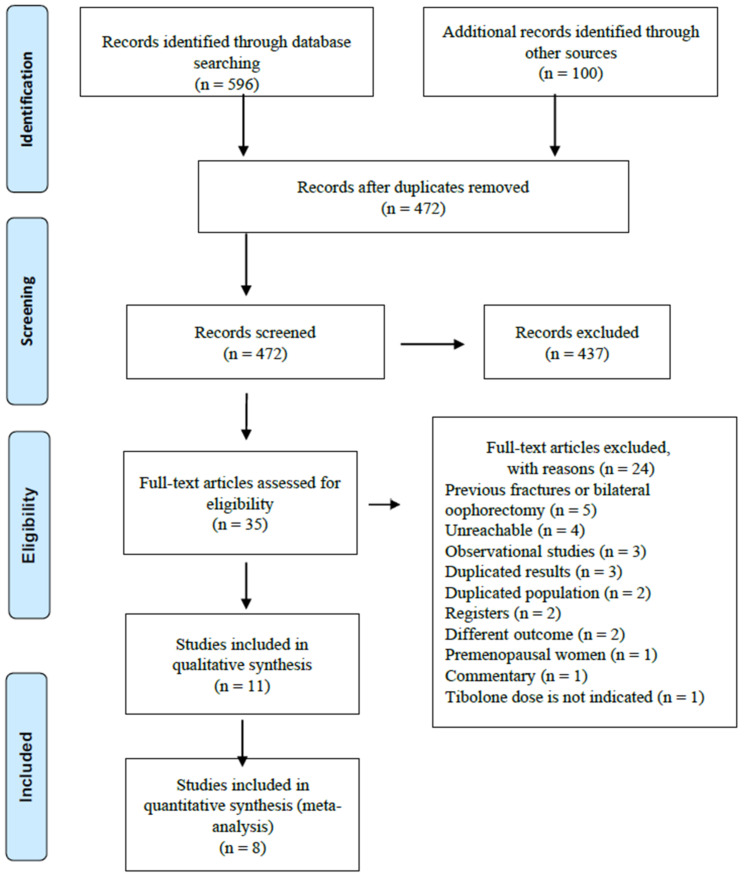
Flow chart of the number of studies included and excluded in the analysis.

**Figure 2 biology-10-00211-f002:**
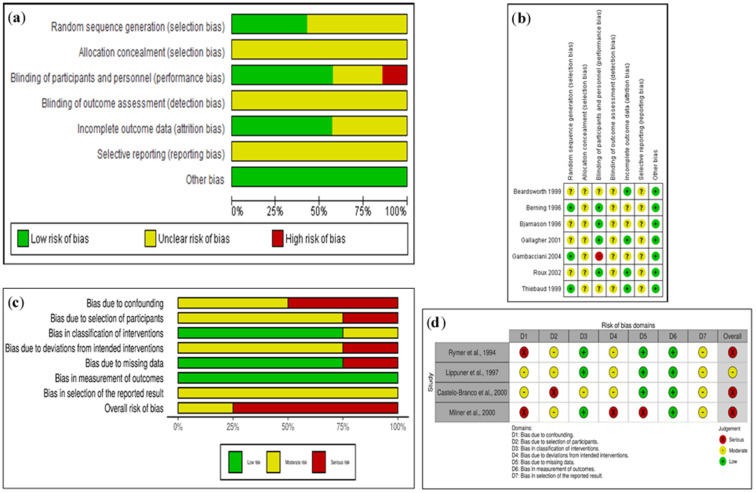
Risk of bias: (**a**) graph and (**b**) summary of randomized studies included in the systematic review (n = 7), low risk “+”, unclear risk “?”; high risk “–”. Risk of bias: (**c**) graph and (**d**) summary of non- and semi-randomized studies included in the systematic review (n = 4).

**Figure 3 biology-10-00211-f003:**
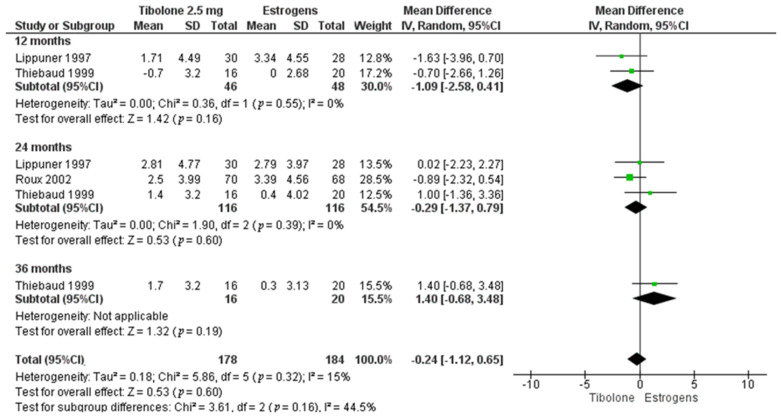
Forest plot of the meta-analysis of mean difference in BMD percentage change from baseline femoral neck through time (12, 24 and 36 months). Hologic densitometer. Tibolone 2.5 mg versus estrogen.

**Figure 4 biology-10-00211-f004:**
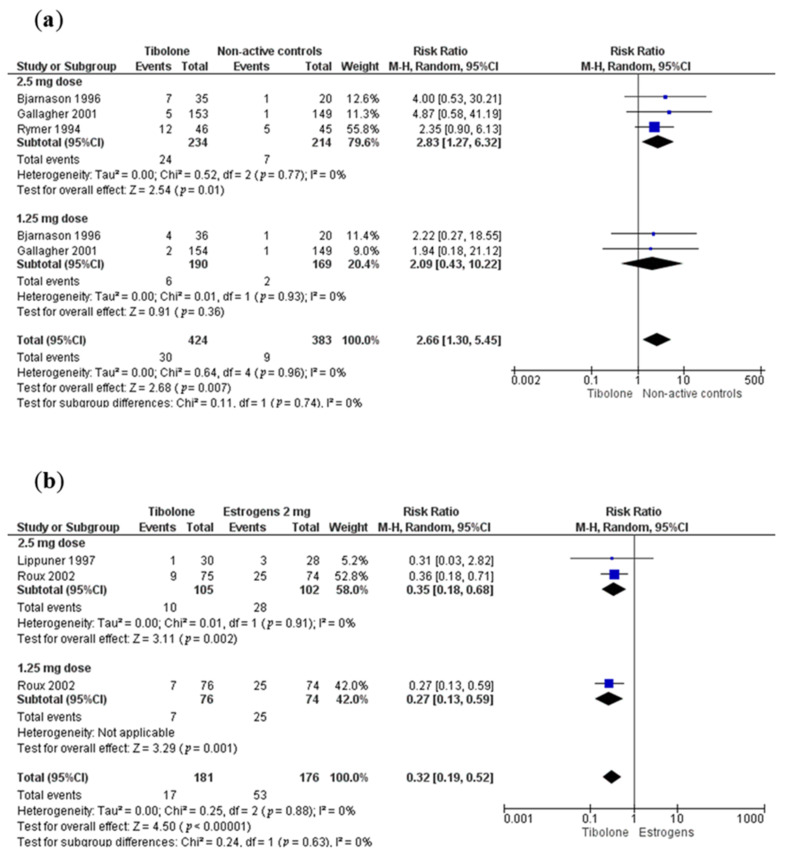
Forest plot of vaginal bleeding. (**a**) Tibolone versus non-active controls, (**b**) Tibolone versus estrogens (2 mg).

**Table 2 biology-10-00211-t002:** Results of the meta-analysis of the mean difference in percentage change from the baseline in bone mineral density. MD; mean difference; 95%CI: 95% confidence interval.

Tibolone vs. Non-Active Controls									
Measurement Site and Follow-Up Months	Tibolone Dose (mg)	Hologic				Lunar			
		No. of Comparisons (References)	MD (95%CI), Participants Random Effect Model	Heterogeneity		No. of Comparisons [References]	MD (95%CI), Participants Random Effect Model	Heterogeneity	
				I^2^ (%)	*p* Value			I^2^ (%)	*p* Value
Lumbar spine 12 m	2.5	4 [14,16,21,24]	2.75 (2.22 to 3.29), 328	0	0.60	2 [18,20]	4.39 (3.34 to 5.44), 173	0	0.35
	1.25	2 [21,24]	3.21 (2.12 to 4.30), 169	34	0.22	1 [20]	3.59 (2.53 to 4.65), 129	-	-
Lumbar spine 24 m	2.5	4 [14,16,21,24]	4.87 (4.16 to 5.57), 328	0	0.81	2 [20,22]	7.35 (2.68 to 12.01), 152	68	0.08
	1.25	2 [21,24]	4.15 (3.27 to 5.03), 169	0	0.32	2 [20,22]	4.84 (3.59 to 6.10), 167	0	0.36
Femoral neck 12 m	2.5	3 [14,16,21]	2.23 (0.89 to 3.57), 287	31	0.24	1 [20]	2.94 (1.65 to 4.23), 126	-	-
	1.25	1 [21]	1.01 (0.12 to 1.96), 128	-	-	1 [20]	2.74 (1.32 to 4.16), 129	-	-
Femoral neck 24 m	2.5	3 [14,16,21]	4.85 (1.55 to 8.15), 287	96	<0.0001	2 [20,22]	4.21 (2.99 to 5.42), 152	0	0.65
	1.25	1 [21]	2.45 (1.48 to 3.42), 128	-	-	2 [20,22]	3.61 (2.41 to 4.80) 167	0	0.87
Tibolone vs. Estrogens									
Lumbar spine 12 m	2.5	3 [16,17,22]	−1.21 (−2.87 to 0.46), 232	67	0.05	1 [18]	−0.30 (−2.82 to 2.22), 46	-	-
24 m		3 [16,17,22]	−0.58, (−3.77 to 2.60), 232	88	0.0002	-	-	-	-
36 m		1 [17]	2.0, (−0.66 to 4.66), 36	-	-	-	-	-	-
Total hip 12 m	2.5	2 [17,22]	−0.81, (−1.58 to −0.03), 174	0	0.52	-	-	-	-
24 m		2 [17,22]	0.12, (−2.28 to 2.53), 174	68	0.08	-	-	-	-
36 m		1 [17]	2.90, (0.62 to 5.18), 36	-	-	-	-	-	-

m, months

## Data Availability

The data that support this study are available from the corresponding author on request.

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
