# Peer review of "Effect of Tibolone on Bone Mineral Density in Postmenopausal Women: Systematic Review and Meta-Analysis"

_biology, 2021, doi:10.3390/biology10030211_

Round 1
Reviewer 1 Report
"Effect of Tibolone on Bone Mineral Density in Postmenopausal 2
Women: Systematic Review and Meta-Analysis" - The manuscript provides a brief overview on the effect of tibolone on BMD of the lumbar spine, femoral neck, and total hip in postmenopausal women by reviewing the quality of some of the selected trials using currently available tools, and further measures the safety of tibolone therapy from the currently available clinical data in the literature.
The basis for the selection of articles is clear and presented well. The analyses were performed appropriately. In conclusion, the authors suggest that Tibolone is useful to prevent the decrease in BMD and highlight four limitations. However, they need to include the strengths of the current literature and recommendations or concerns for future studies.
Minor:
The manuscript has to be improved for readability.
Author Response
Comment 1:
“The manuscript provides a brief overview on the effect of tibolone on BMD of the lumbar spine, femoral neck, and total hip in postmenopausal women by reviewing the quality of some of the selected trials using currently available tools, and further measures the safety of tibolone therapy from the currently available clinical data in the literature.
The basis for the selection of articles is clear and presented well. The analyses were performed appropriately. In conclusion, the authors suggest that tibolone is useful to prevent the decrease in BMD and highlight four limitations. However, they need to include the strengths of the current literature and recommendations or concerns for future studies.”
Answer 1:
The requested information was included at the end of the discussion. The text is in red and was highlighted in yellow.
Comment 2:
“The manuscript has to be improved for readability.”
Answer 2:
The manuscript was edited. The edition result was highlighted in yellow.

Reviewer 2 Report
The manuscript entitled, Effect of Tibolone on Bone Mineral Density in Postmenopausal Women: Systematic Review and Meta-Analysis, is a systemic review with meta-analyses to investigate the how the low risk hormone replacement therapy, tibolone, impacts postmenopausal osteoporosis.
This manuscript reports on a topic that is likely to be of interest to the readers and could have a significant impact on the field. However, in its current form the manuscript should be significantly revised.
Specifically, while the authors do an acceptable job of comparing and contrasting tibolone to estrogen (and SERMs), the authors much include some other comparisons to more traditional osteoporosis drugs (bisphosphonates, denosumab, PTH/PTHrP, etc). This is incredibly important as these drugs are classically the first line of defense to combat skeletal fragility associated with postmenopausal osteoporosis. Therefore, it stands to reason that comparing tibolone to these other drugs would be beneficial and informative.
Author Response
Comment:
“This manuscript reports on a topic that is likely to be of interest to the readers and could have a significant impact on the field. However, in its current form the manuscript should be significantly revised.
Specifically, while the authors do an acceptable job of comparing and contrasting tibolone to estrogen (and SERMs), the authors mucho include some others comparisons to more traditional osteoporosis drugs. This is incredibly important as these drugs are classicallly the first line of defense to combat skeletal fragility associated with postmenopausal osteoporosis. Therefore, it stands to reason that comparing tibolone to theses other drugs would be benefical and informative.”
Answer:
We thank the reviewer for this comment. In this sense, we want to clarify that the hormone therapy is the first-line treatment to alleviate climacteric symptoms in postmenopause. In addition, tibolone offers benefits compared to estrogens, it has fewer adverse effects, and both medications prevent bone loss. Therefore, it is possible to prevent the development of osteoporosis and avoid the intervention of pharmacological options (antiresorptive medications or stimulators of bone formation) to treat this disease.
This systematic review and meta-analysis focuses on the prevention of decreased bone mineral density. The population of the trials included in this review are women without osteoporosis and previous fractures, for this reason comparisons with osteoporosis treatments were not carried out. It is indicated that the evidence from this research cannot extrapolate to this population.
As your suggestions, we added the information about this topic in the Introduction, study selection and Discussion sections. The text is in red and highlighted in yellow.

Round 2
Reviewer 2 Report
The revised manuscript is acceptable.